# EWSR1::ATF1 Orchestrates the Clear Cell Sarcoma Transcriptome in Human Tumors and a Mouse Genetic Model

**DOI:** 10.3390/cancers15245750

**Published:** 2023-12-08

**Authors:** Benjamin B. Ozenberger, Li Li, Emily R. Wilson, Alexander J. Lazar, Jared J. Barrott, Kevin B. Jones

**Affiliations:** 1Department of Oncological Sciences, University of Utah School of Medicine, Salt Lake City, UT 84132, USA; bozenber@gmail.com (B.B.O.); li.li@hci.utah.edu (L.L.); emily.r.wilson@hci.utah.edu (E.R.W.); 2Department of Orthopaedics, University of Utah School of Medicine, Salt Lake City, UT 84132, USA; 3Department of Pathology, Genomic Medicine and Dermatology, The University of Texas MD Anderson Cancer Center, Houston, TX 77030, USA; alazar@mdanderson.org; 4Department of Biology, Brigham Young University, Provo, UT 84602, USA; jared_barrott@byu.edu

**Keywords:** epigenetics, chromatin, comparative genomics, transcription factor

## Abstract

**Simple Summary:**

Many cancers, especially those most common in adolescents and young adults, are initiated by a single change in the DNA sequence of a gene. The particular change creates a new gene through the fusing of the beginning of one gene to the end of a second gene. This new fusion gene will be expressed as a fusion protein that combines the functions of the two parent genes, but in new ways. Because many of the cancers that are driven by these fusion genes are rare, study of the precise function of the fusion proteins can be difficult. This article reports on the study of a particular fusion protein that drives the rare, but deadly, pediatric cancer clear cell sarcoma. The fusion protein was expressed in mice, leading to the formation of similar tumors to those it causes in humans. This permitted the study of fusion protein function in these tumors.

**Abstract:**

Clear cell sarcoma (CCS) is a rare, aggressive malignancy that most frequently arises in the soft tissues of the extremities. It is defined and driven by expression of one member of a family of related translocation-generated fusion oncogenes, the most common of which is *EWSR1::ATF1*. The EWSR1::ATF1 fusion oncoprotein reprograms transcription. However, the binding distribution of EWSR1::ATF1 across the genome and its target genes remain unclear. Here, we interrogated the genomic distribution of V5-tagged EWSR1::ATF1 in tumors it had induced upon expression in mice that also recapitulated the transcriptome of human CCS. ChIP-sequencing of V5-EWSR1::ATF1 identified previously unreported motifs including the AP1 motif and motif comprised of TGA repeats that resemble GGAA-repeating microsatellites bound by EWSR1::FLI1 in Ewing sarcoma. ChIP-sequencing of H3K27ac identified super enhancers in the mouse model and human contexts of CCS, which showed a shared super enhancer structure that associates with activated genes.

## 1. Introduction

Epigenetic reprogramming is an emerging hallmark of cancer, which involves the aberration of nuclear machinery that regulates gene expression via genome accessibility, enhancer activity, and secondary histone modifications [1]. Some malignancies, including subtypes of leukemia and sarcoma, are initiated by singular genetic events that generate pathognomonic fusion genes through chromosomal translocation [2]. Many of these fusion genes express fusion proteins that function as transcription factors (TFs) that profoundly reprogram transcription, driving oncogenesis. Many such fusion oncoproteins remain poorly characterized with regard to their epigenetic mechanisms of gene regulation, genome-wide chromatin distribution, and target genes.

Clear cell sarcoma (CCS), previously known as malignant melanoma of soft parts, is a soft-tissue sarcoma that arises in muscle compartments, tendons, or aponeuroses, most frequently in the extremities [3]. The 5-year and 10-year survival rates for patients diagnosed with CCS are 50% and 38%, respectively [4]. The definitional chromosomal translocation for CCS fuses *EWSR1* on chromosome 22q12 to *ATF1* on chromosome 12q13. The *EWSR1::ATF1* fusion oncogene (hereafter referred to as *EA1*) is identified in greater than ninety percent of CCS cases. *EA1* expression can independently initiate clear cell sarcomagenesis, demonstrated by multiple mouse genetic models [5,6]. In rare cases of CCS, alternative fusion genes are identified, involving related genes, such as *EWSR1::CREB1* or *EWSR1::CREM* [7,8]. The family of *EWSR1/FUS::CREB* translocation-associated tumors also includes angiomatoid fibrous histiocytoma, gastrointestinal clear cell sarcoma, pulmonary myxoid sarcoma, and a couple of carcinomas [9,10,11]. *EWSR1* is involved in many other translocations across sarcoma subtypes, including *EWSR1::FLI1* in Ewing sarcoma [12].

EWSR1 and other members of the FET family of proteins, FUS and TAF15, are RNA-binding proteins that are thought to function in splicing, but also have the distinct capacity to phase separately, which has prompted recent exploration of their roles in phase separated transcription hubs [13,14]. ATF1 is a member of the CREB/ATF family of TFs, along with CREB1 and CREM [15]. Importantly, the *EA1* fusion gene retains the bZIP DNA-binding domain of *ATF1* yet excludes the cyclic AMP (cAMP) response element, which has led to the hypothesis that EA1 is a cAMP-independent, hyperactive transcription factor for ATF1 target genes [16,17]. Moreover, the retained EWSR1 domain is presumed to be transcription-modulating due to its prion-like domain, which is shown to be crucial in the reprogramming that drives Ewing sarcoma [18,19,20].

CCS accounts for less than one percent of soft-tissue sarcomas and fewer than 100 cases are reported annually in the United States [3]. This scarcity of human cases was addressed by the development of genetically engineered mouse models that recapitulate the histologic features and transcriptome of human CCS [5]. Another challenge to the epigenomic characterization of CCS has been the lack of high-quality antibodies against the fusion specifically or even the amino terminal half of EWSR1. A prior investigation attempted to discern the genomic distribution of EA1 across chromatin by the intersection of chromatin immunoprecipitation sequencing (ChIP-seq) using antibodies against amino terminal EWSR1 and carboxy terminal ATF1 [21]. While this offered a closer look into the CCS epigenome than previously afforded, the analyses were still limited by the poor quality of the available EWSR1 antibodies and the artifacts created by contamination of each ChIP-seq experiment with endogenous EWSR1 or ATF1 distribution across chromatin. Such challenges with poor-quality antibodies for fusion oncoproteins have previously been shouldered by CRISPR/Cas9 efforts to tag a native fusion in a cancer cell line [21,22]. However, even such experiments carry the potential for an artifact as the tag may alter the oncogenicity of the fusion in unknown ways. We therefore determined to generate tumors from a tagged version of EA1 in the mouse, then interrogate these tumors to discover what could be learned about the human CCS epigenome.

## 2. Materials and Methods

### 2.1. Mouse Genetic Model of CCS (mCCS)

All animal experiments were performed under the approval of the University of Utah Institutional Animal Care and Use Committee and in accordance with international law and humane principles. The protocol under which this mouse strain was generated, these mouse tumors were grown, and euthanasia criteria and timing were described was protocol #16-10012, approved on 28 November 2016. The mouse strain background was 129/SvJ and C57Bl/6 as previously described [5]. Expression of type 1 *hEWSR1::ATF1* (exon 8::exon 4) from the *Rosa26* promoter was conditional upon hindlimb injection of purified TATCre, a soluble form of Cre-recombinase that localizes to nuclei through the TAT epitope. TATCre injections into mice were performed as previously described [5]. Mouse tumor growth progression was measured using calipers and calculated using an ellipsoidal formula: volume = (length × width × width) × 0.5. Mice were monitored for signs of distress that indicated euthanasia via CO_2_ followed by cervical dislocation, in accordance with a veterinarian-approved protocol. Tumors were surgically removed using sterilized scalpels and forceps, placed into 1.5 mL tubes, and immediately frozen with liquid N_2_.

### 2.2. Immunochemistry of Mouse Model Tumors

Four mouse model tumors were powderized using a cryogenic freezer mill. Frozen tumor powder was lysed using a RIPA buffer (Alfa Aesar, Haverhill, MA, UA; J62524) with a protease inhibitor cocktail (Sigma-Aldrich, St. Louis, MO, USA; P8340). Immunoprecipitation on lysate was performed using an anti-V5 pre-conjugated agarose slurry (Sigma-Aldrich A7345). An immunoblot to confirm expression of V5-tagged *EA1* was performed using an anti-V5 (Bethyl Laboratories, Inc, Montgomery, TX, USA; A190-120A 1:2000) or anti-EWSR1 (Lifespan Biosciences, Lynnwood, WA, USA; LS-B7255 1:5000) primary antibody followed by a Protein A secondary antibody (Bio-Rad, Hercules, CA, USA; 1706522 1:5000) on the immunoprecipitated and input samples. A ladder (Thermo Fisher Scientific, Waltham, MA, USA; 26619) was run in parallel to measure size of proteins in kDa. The original immunoblot figures can be found in Appendix A.

### 2.3. Human Models of CCS

The SU-CCS-1 human cell line was purchased from ATCC (CRL-2971). SU-CCS-1 cells were cultured in RPMI-1640 (Thermo Fisher 11875093) supplemented with 10% fetal bovine serum (Thermo Fisher 26140079). Cell cultures were incubated at 37 °C and 5% CO_2_ in non-treated flasks. Five human CCS samples were flash-frozen and shared to us from Dr. Lazar (MD Anderson Cancer Center). These human specimens were deidentified, by our use of them, but were collected with consent and IRB approval at MD Anderson Cancer Center.

### 2.4. Knockdown of EA1 by siRNA Transfection

siRNA targeting the *EA1* fusion gene transcript (Thermo Fisher s1696) or siRNA negative control (Thermo Fisher 4404021) was transfected in SU-CCS-1 cells for 48 h. Transfection was accomplished using 5 pmol of siRNA and 2 µL of lipofectamine (Thermo Fisher 13778-075) in 1.2 mL of Opti-MEM media (Thermo Fisher 31985070) per well of a 12-well non-treated plate.

### 2.5. RNA-Sequencing Acquisition and Data Analysis

RNA purification began by suspending the sample pellet in TRIzol (Ambion, Inc., Life Technologies, Carlsbad, CA, USA; 15596018) followed by chloroform to a percentage by volume of 16.67%. Then, RNA from suspended lysate was purified and eluted in water using a kit (Zymo Research, Irvine, CA, USA; RNA Clean & Concentrator R1018). RNA-seq libraries, constructed with a TruSeq Stranded mRNA Library Prep kit with Unique Dual Index Primers (Illumina, San Diego, CA, USA), were sequenced on an Illumina NovaSeq instrument using the 2 × 50 bp protocol, at approximately 25 million reads per sample depth. Reads were aligned to mm10 or hg38 using BWA-MEM (version 0.7.10-r789) [23]. FeatureCount (version 1.6.3) generated count matrices and DESeq2 3.11 generated a differential expression analysis [24,25]. Differentially expressed genes with a false-discovery rate of <0.05 were regarded as statistically significant. The regularized log (rlog) counts were used for visualization by heatmap.

For comparative transcriptomics, we collected data from 5 human CCS tumors, 5 human skeletal muscle samples, 10 human MRT (malignant rhabdoid tumor) samples, 4 mouse CCS tumors, and 3 mouse skeletal muscle samples [26]. Rlog values were generated with DESeq2 [25]. The batch effect was removed by ComBat_seq between batches of experiments [27]. There are 10,772 genes with homologues between a mouse and human in the merged dataset. The PCA was performed on the top 10,000 most variable genes, and the PCA plot depicted PC1 and PC2. Significant, differentially expressed genes were identified with the cut-offs of an adjusted *p*-value of less than 0.05 in all 3 datasets and an absolute log_2_(fold change) value of greater than 1 in mCCS and hCCS datasets. A KEGG pathway enrichment analysis was performed using Enrichr [28].

### 2.6. ChIP-Sequencing of Mouse Model Tumors

Mouse model tumors were dissected, snap-frozen, and powderized using a cryogenic freezer mill. Frozen tumor powder was suspended in 1% formaldehyde in PBS for cross-linking followed by the addition of glycine to a final concentration of 125 mM. Nuclei were isolated by suspending and douncing in a cell lysis buffer containing 10 mM Tris-HCl (pH 8.0), 0.5% NP-40, and 10 mM NaCl. Nuclear pellets were lysed using a nuclear lysis buffer containing 1% SDS, 50 mM Tris-HCl (pH 8.0), 10 mM EDTA, and 100 mM NaCl. Nuclear lysates were sonicated using a 4 °C water bath sonicator (Diagenode Bioruptor Pico) for 8 cycles of 30 s on and 30 s off. Sonicated chromatin was incubated with 5 µg of a primary antibody overnight with rotation at 4 °C (5 samples with Bethyl A190-120A anti-V5, 5 samples with Abcam ab5131 (Abcam, Cambridge, UK) anti-RNAPOL2, or 4 samples with Abcam ab4279 anti-H3K27ac). Samples were then incubated with 100 µL DynaBeads for 4.5 h with rotation at 4 °C. Conjugated DynaBeads were eluted using a buffer containing 1% SDS, 10 mM Tris-HCl (pH 8.0), 5 mM EDTA, and 150 mM NaCl and by incubating at 65 °C for 9 h. Immunoprecipitated DNA was purified and eluted in water using a kit (Zymo Research DNA Clean & Concentrator D4034).

### 2.7. ChIP-Sequencing of Human CCS Cell Lines

Anti-EWSR1 and anti-ATF1 ChIP-seq was performed on SU-CCS-1 and DTC-1 cell lines and processed as reported by Möller et al. [29]. An intersectional analysis was performed by doing iterations of the bedtools intersect on processed bed files [30]. The cell pellet (approximately 5 million cells) was prepared for ChIP-seq as described for mouse model tumors other than the initial processing with a freezer mill. Chromatin immunoprecipitation was performed using 5 µg of an anti-H3K27ac antibody (Abcam ab4729) overnight at 4 °C.

### 2.8. ChIP-Sequencing Analysis

Libraries constructed with a NEBNext ChIP-seq Library Prep kit with Unique Dual Index Primers (New England Biolabs, Ipswich, MA, USA) were sequenced on an Illumina NovaSeq, using the 2 × 50 bp protocol to approximately 30 million reads per sample depth. ChIP-seq reads were aligned to the mm10 or hg38 using BWA-MEM (version 0.7.10-r789) [23]. MACS2 version 2.2.6 called peaks with the parameters “callpeak-B–SPMR–*p* value = 1 × 10^−10^–mfold 15 100” using ChIP input as the background [31]. MACS normalized bedgraphs, which were converted to bigwig files. The –broad option in MACS2 was used for anti-H3K27ac ChIP-seq. Peaks were filtered to remove ENCODE denylisted region peaks [32]. Samtools rmdup removed duplicate reads [33]. Multiple replicates were merged into bigwig enrichment files in an average manner, then normalized for read depth. The ChIPseeker annotatePeak tool with “tssRegion” set to “c(−3000, 3000)” calculated genomic distributions [34]. Heatmaps and profile plots of ChIP-seq enrichment associated with genomic regions were generated with plotHeatmap after calculating scores per genome region and an intermediate file was prepared with computeMatrix of deepTools 3.3.2 [35]. The profile plots of peak distribution and intersection between ChIP-seq datasets were generated using ChIPpeakAnno version 3.22.0 [36]. Genomic tracks depicting ChIP-seq and RNA-seq enrichments (reads per million) were captured using IGV version 2.4.14.

EA1 binding sites were ascribed relative to their nearest two genes within 500 kb upstream or downstream, annotated in mm10 or hg38, using GREAT [37]. Identification of recurrent motifs in EA1 binding sites was accomplished using MEME with the parameter “any number of repeats” [38]. We developed an algorithm to measure the frequency of repeated TGA or the reverse complement TCA in EA1 binding sites. The algorithm scanned the DNA sequence with a window of 3 base pairs (bp) of length and step of 1 bp, searching for the pattern TGA or TCA. The algorithm allowed for a 5 bp gap tolerance and a threshold of 4 occurrences. When the gap between two patterns was more than 5 bp, the algorithm started over. When the accumulating occurrence of the pattern is greater than the threshold of 4, the algorithm will record the event as one repeating sequence.

From the anti-H3K27ac ChIP-seq data, super enhancers were called by ROSE (Rank Ordering of Super Enhancers) [39,40]. Briefly, peaks were called by MACS2; then, peaks within 12.5 kb were stitched together into larger regions. The SE signal of each of these regions is determined by the total reads normalized by the input. Syntenic SEs were identified by using the UCSC LiftOver tool. Mouse SEs were translated to hg38 loci then compared to human SEs, and human SEs were translated to mm10 loci then compared to mouse SEs. Anti-H3K27ac HiChIP-seq was performed on the SU-CCS-1 human cell line and analyzed as reported by Möller et al. [29].

For additional graphical representations, we utilized R programming throughout this study. The bespoke algorithm developed for motif scanning was made publicly accessible via our GitHub repository: GitHub Link. Given that the method is a specialized solution rather than a full-fledged package, it has been shared within an existing repository rather than establishing a separate one.

### 2.9. Tumor Processing for Histology

Tumor samples were fixed in 4% formaldehyde, dehydrated in serial ethanol solutions to xylene, then embedded in paraffin. Additionally, 4 μm sections were cut with a microtome and mounted on slides. Slides were deparaffinized with xylene, rehydrated with serial ethanol, and stained with hematoxylin and eosin for light microscopy viewing.

## 3. Results

### 3.1. Conditional Expression of V5-Tagged EWSR1::ATF1 (EA1) Drives Tumorigenesis, Which Recapitulates the Human CCS Transcriptome

A V5 tag was added to the complementary DNA (cDNA) for the *EA1* coding sequence, then inserted by homologous recombination into the *Rosa26* locus, a modestly expressed mouse pseudogene. It was separated from the *Rosa26* promoter by a *loxP*-flanked stop cassette, which would be removed upon exposure to the Cre-recombinase enzyme. The cDNA was followed by enhanced green fluorescent protein on an internal ribosomal entry site for tandem marker expression with the fusion (Figure 1A). Mouse tumors were palpable within 1 month of induction with 100% penetrance, and growth rates were comparable between mouse model tumors expressing V5-tagged or the previously validated, untagged EA1 (Figure 1B). Expression of *EA1* initiated by TATCre protein injection into limbs of 4-week-old mice generated tumors that were indistinguishable from prior tumors generated with this technique with an untagged fusion (Figure 1C,D). These observations offered confidence that introduction of the tag had not altered the tumorigenesis program driven by EA1. Protein-level expression of V5-tagged EA1 (approximately 80 kDa) was verified by anti-V5 immunoprecipitation followed by an anti-V5 immunoblot. An anti-EWSR1 immunoblot identified the tagged fusion V5-EA1 as well as the full-length endogenous EWSR1 at approximately 95 kDa (Figure 1E).

To test the transcriptome generated in these tumors against the human CCS transcriptome, we performed bulk RNA-sequencing (RNA-seq) on four mouse model tumors and five human CCS frozen samples. Sequencing results were compared to RNA-seq results from three mouse skeletal muscle samples and five human skeletal muscle samples, respectively. Although muscle is not considered to be a specific cell of origin for CCS, it is the typical host tissue for most human CCS tumors and is readily available from both species, making it a reasonable comparison tissue for controls. The principal component analysis (PCA) showed distinct clustering between CCS and muscle. However, PCA showed unique clusters of gene expression profiles between CCS contexts, suggesting species-specific signatures (Figure 1F).

MRT was recapitulated in genetically engineered mice in which recombinase induced genetic loss of *Smarcb1*, the consistent genetic alteration known to initiate the malignancy [41,42]. RNA-seq followed by a parallel PCA analysis was performed on mouse MRTs and human MRT samples. These MRT sample groups clustered together individually and clustered adjacent to one another, supporting the idea that mouse modeling recapitulates transcription signatures across these relatively simple genome cancers (Figure 1F).

### 3.2. Transcriptomic Analysis of Mouse and Human CCS Identified Shared Differentially Expressed Genes Involved in Proliferation

The RNA-seq analysis revealed differentially expressed genes (DEGs) in the mouse and human CCS tumors compared to normal skeletal muscle of their respective species. A total of 4444 upregulated genes (log_2_ transformed fold-change > 1, adjusted *p*-value < 0.05) and 3630 downregulated genes (log_2_FC < −1, padj < 0.05) were identified in mouse CCS tumors (Figure 2A). A total of 4754 upregulated genes (log_2_FC > 1, padj < 0.05) and 3847 downregulated genes (log_2_FC < −1, padj < 0.05) were identified in human CCS tumors (Figure 2B). A total of 1378 upregulated genes and 1085 downregulated genes were shared between mouse and human CCS tumors.

In the human SU-CCS-1 cell line, RNAi knockdown of *EA1* followed by a gene expression analysis demonstrated other functionally relevant DEGs. Contrasted to the analyses performed above, this analysis identified DEGs as a result of alterations in *EA1* expression, rather than differentially associated with either the tumor or muscle cell state. A total of 2451 activated genes (log_2_FC > 0, padj < 0.05) and 2160 repressed genes (log_2_FC < 0, padj < 0.05) were identified with the differential expression analysis comparing RNAi control to RNAi knockdown of *EA1* (Figure 2C).

The shared intersection of all three datasets constitutes a comprehensive list of DEGs that may be critical in CCS initiation or progression. A total of 2253 upregulated genes and 1655 downregulated genes were identified across two or more CCS datasets (Figure 2D). A total of 432 upregulated genes and 194 downregulated genes were shared across all three CCS datasets. Interesting upregulated genes shared across all datasets included *CREM*, *FOS*, *JUNB*, *NONO*, *RUNX1*, and *RUNX2*. Upregulated genes were significantly enriched for pathways involved in cell cycle and DNA replication (Figure 2E). These pathways suggest that EA1 activates proliferation. Downregulated genes were significantly enriched for pathways involved in insulin signaling, oxidative phosphorylation, the TCA cycle, and PPAR signaling (Figure 2F). The downregulation of canonical metabolic pathways suggests repressed genes may play a role in switching from a normal cellular metabolism to oncometabolism [43].

### 3.3. Genomic Binding of EWSR1::ATF1 Shows Predilection of Distal Regions and Correlation with RNAPOL2 and H3K27ac Signatures

The EA1 fusion oncoprotein is assumed to function as a reprogramming transcription factor because it retains the nuclear localization signal and the DNA-binding domain from ATF1. However, the genomic binding pattern of EA1 is largely unknown. We first tested the hypothesis that EA1 binds similarly to wildtype ATF1. Chromatin immunoprecipitation followed by next-generation sequencing (ChIP-seq) identified genomic loci that are bound by EA1. Using the TATCre-inducible V5-tagged *EA1* mouse model, anti-V5 ChIP precipitated EA1-bound loci for sequencing (*n* = 5). A total of 21,237 EA1 binding sites were identified across the mouse genome. The genomic distribution of EA1 ChIP-seq peaks revealed that the fusion oncoprotein binds promoter regions within +/− 3 kb of a gene’s transcription start site (34.7%), intragenic regions (36.5%), and intergenic regions (28.8%) (Figure 3A).

EA1 binding was previously identified in human CCS cell lines, SU-CCS-1 and DTC1, as mentioned above, using the intersectional analysis of ChIP-seq for either terminus of the fusion oncoprotein [29]. A total of 2314 consensus binding sites were identified by performing the intersectional analysis of these anti-ATF1 and anti-EWSR1 ChIP-seq datasets from both cell lines that were made publicly available. The substantially smaller number of binding sites identified may be due to the difficulty of performing ChIP using an antibody for EWSR1. The fraction of promoter-associated binding sites (18.0%) in the human context was particularly less than the fraction identified in mouse model tumors (34.7%) (Figure 3B). To account for the threshold stringency artifact, the analysis of mouse tumor binding sites was performed on the 2314 most significantly enriched binding sites. Genomic distribution of these high-stringency binding sites was comparable to the analysis of all 21,237 binding sites, suggesting EA1 can strongly bind at promoter, intragenic, or intergenic regions in mCCS (Figure 3C).

ATF/CREB factors regulate target genes directly by binding promoter regions [44]. A total of 7370 (34.7%) EA1 binding sites demarcated promoters of genes that may be directly regulated by fusion protein binding similar to wildtype ATF/CREB factor binding. Previously published ChIP-seq of wildtype ATF1 in the HepG2 hepatocellular carcinoma cell line identified 69.2% of 8863 ATF1 binding sites in promoters (Figure 3D) [45]. A significantly smaller fraction of promoter binding for EA1 supports the idea that the fused portion of EWSR1 confers the capacity to distribute to intragenic and intergenic (distal) regions more avidly than the native ATF1 alone.

EA1 binding sites in mCCS were ascribed to the gene with the nearest transcription start site (TSS) and were represented by a significance score (Figure 3E, left) and by a count (Figure 3E, right). Promoter-associated binding sites (+/−3 kb of TSS) included binding sites that directly overlapped the TSS, represented at the *y*-axis value of 0. The most significantly enriched promoter-associated binding site directly overlapped the TSS of one isoform of *Crem*. *Crem* was significantly upregulated in all three RNA-seq datasets (padj < 0.03). CREM is also a member of the ATF/CREB family of transcription factors and the EWSR1::CREM fusion has been reported in a few cases of CCS [8,46].

ChIP-seq experiments using antibodies for H3K27ac (*n* = 4) and phosphorylated RNA polymerase II (RNAPOL2) (*n* = 5) were performed in parallel to EA1. The distribution of these factors, whose presence indicates actively transcribed regions and enhancers, overlapped with EA1 binding (*n* = 10,283) (Figure 3F). Enrichment heatmaps representing the fractions of binding sites shared between the three datasets showed substantial correlation between ChIP-seq enrichments across loci, even when binding at some loci did not meet the threshold to be called as peaks (Figure 3G). H3K27ac enrichment showed enhancer-like distributions in clusters #1, #3, and #5, where EA1 occupied the center of these loci, flanked by adjacent/surrounding H3K27ac enrichment. Cluster #1 demonstrated strongly enriched, distal EA1 peaks with flanking H3K27ac enrichment and weaker RNAPOL2 enrichment, versus reduced EA1 enrichment that coincided with high RNAPOL2 enrichment and promoter-type single peak enrichment for H3K27ac.

### 3.4. EWSR1::ATF1 Binds the Canonical ATF/CREB Motif at Promoters and Novel Variant Motifs at Distal Regions

We hypothesized that EA1 binds at wildtype ATF1 binding sites and, more specifically, at the canonical ATF/CREB binding motif because the fusion protein retains the ATF1 bZIP DNA-binding domain. Wildtype ATF/CREB factors are known to dimerize at the palindromic, 8-base-pair motif (TGACGTCA) [47]. The analysis of recurrent motifs in the EA1 ChIP-seq dataset demonstrated high occurrence of this expected ATF/CREB motif (Figure 4A). The interior (CG) of the canonical motif appeared to be flexible, with frequent occurrences of motifs such as TGACATCA as well. Another recurrent motif across EA1 binding sites matched a known AP1 motif (TGAGTCA). AP1 is a transcription factor heterodimer with JUN and FOS proteins bound typically to distal enhancer regions [48,49]. Lastly, a repetitive sequence composed of TGA repeats was identified as a recurrent motif that may represent novel binding sites for EA1.

Approximately 50% of binding sites that contain the canonical ATF/CREB motif were promoter-associated binding sites, compared to only about 20% of binding sites that contain variant motifs, TGANNTCA or TGAGTCA (Figure 4B).

A motif analysis of EA1 ChIP-seq in human CCS cell lines also revealed high occurrences of the canonical ATF/CREB motif (Figure 4C). The fifth base pair of the motif appeared indiscriminate, corroborating flexibility of the center of the motif observed in mCCS EA1 binding sites. TGA repeats were also identified as a recurrent sequence motif in the human EA1 binding sites. The AP1 motif (TGAGTCA) was not identified as a significant, recurrent motif.

In Ewing sarcoma, the EWSR1::FLI1 fusion oncoprotein binds GGAA repeats in microsatellite regions to generate novel enhancers [50]. These EWSR1::FLI1-bound microsatellites contain between 15 and 25 GGAA repeats, suggesting multiple EWSR1::FLI1 proteins may co-bind in multi-valent fashion at each microsatellite [51]. For EA1, we identified strings of 3-base-pair repeats at the center of binding sites that are primarily distal in mouse and human contexts of CCS (Figure 4D). These strings of TGA repeats represent a novel motif at enhancers that wildtype ATF1 does not bind. They may also suggest that EWSR1 is contributing a multi-valent binding capacity to the EA1 fusion oncoprotein, similar to its role in Ewing sarcoma [19].

Similar to Figure 3G, enrichment of EA1, RNAPOL2, and H3K27ac was correlated when clustered by the motif within the EA1 binding sites (Figure 4E). In cluster #1, EA1 binding sites that contained the canonical TGACGTCA motif associated with a uniform distribution of H3K27ac enrichment and were more likely to be promoter-associated. In clusters #2, #3, and #4, EA1 binding sites contained novel variant motifs, were relatively stronger in enrichment, were associated with enhancer-pattern/flanking H3K27ac enrichment, and were more likely to be distal.

In mCCS, the 12th strongest EA1 binding site (out of 21,237) was located approximately 10 kb downstream of *Fam43a* and contained four strings of TGA repeats at its central 100 base pairs (Figure 4F). *Fam43a* is significantly upregulated in all three RNA-seq datasets, suggesting that EA1 may transcriptionally activate the gene from this nearby regulatory region. In human CCS cell lines, the 39th strongest EA1 binding site (out of 2314) was located approximately 5 kb downstream of *TCEAL8* and contained a string of twelve TGA repeats (Figure 4G). *TCEAL8* is also significantly expressed in all three RNA-seq datasets. Both peaks contain TGA repeats, are bound strongly by EA1, and represent enhancer regions adjacent to transactivated DEGs.

### 3.5. Promoter-Associated EA1 Binding Sites Promoted Transcription of Target Genes

To investigate if EA1 is an activating or repressing transcription factor, combinatorial analyses of ChIP-seq and RNA-seq data identified DEGs that are targets of EA1 across mouse and human contexts of CCS. In mCCS, substantial overlap was observed between promoter-associated EA1, RNAPOL2, and H3K27ac signatures, indicating transcriptional activation of these genes (Figure 5A). Strong correlation was observed between EA1, H3K27ac, and RNAPOL2 enrichment centered at all TSSs (Figure 5B).

EA1 binding sites that occur within a gene’s promoter may directly regulate transcription of that target gene. A comparative analysis can be performed on DEGs identified in Figure 2. In total, 188/432 (44%) upregulated genes have EA1 binding in their promoter regions in the mouse model tumors (Figure 5C). In total, 68/194 (35%) downregulated genes have EA1 promoter binding. These data indicate that EA1 can associate with upregulated or downregulated direct target genes. Similar fractions were observed for DEGs shared in any two of the RNA-seq datasets. Much smaller fractions of DEGs contained EA1 promoter binding in the human cell line data (28/432 upregulated genes and 1/194 downregulated genes) due to the identification of only 2314 binding sites (Figure 3B).

A total of 7370 EA1 binding sites occurred at the promoter regions of 5977 unique genes, which were more likely to be transcriptionally activated than repressed. In the mCCS RNA-seq dataset, 1002 of these target genes were significantly upregulated (log_2_FC > 2, padj < 0.05) and 723 target genes were downregulated (log_2_FC < −2, padj < 0.05). Furthermore, promoter-associated EA1 binding sites at upregulated genes were significantly more enriched (average score of 387.0; >1 log_2_FC) than EA1 enrichment at unregulated genes (average score of 272.4; log_2_FC between 1 and −1) or downregulated genes (average score of 213.1; <−1 log_2_FC) (Figure 5D). These observations support that EA1 binds more strongly at upregulated target genes and functions more likely as a transcriptional activator than as a repressor.

To analyze the transcriptional activity of EA1 in human CCS, 416 EA1 binding sites in the human cell lines bound to the promoter regions of 344 unique genes. Again, these genes were more likely to be transcriptionally active. From the SU-CCS-1 RNA-seq dataset evaluating *EA1* knockdown, 81/344 (23.5%) genes that contain promoter-associated EA1 had log_2_FC values greater than 1, compared to 8/344 (2.3%) that had log_2_FC values less than −1. More investigation is needed to determine if the promoters of transcriptionally inactive genes may be functioning as enhancers that loop to other promoters (termed E-promoters) [52].

In mCCS, the third strongest EA1 binding site (out of 21,237) occurred at the promoter of *Crem* (Figure 5E). *Crem* expression was most significant at the furthest downstream exons, suggesting that there may be an alternative promoter and a unique isoform as seen in human germ cells [53]. In human CCS cell lines, EA1 binding occurred at the promoter of *NFIL3* (Figure 5F). Human *CREM* and mouse *Nfil3* also contained promoter-associated EA1 binding, although ChIP-seq enrichment was weaker than in the other species. *CREM* and *NFIL3* are significant, upregulated genes in all three RNA-seq datasets.

### 3.6. Distal EA1 Binding Sites Demarcated Homologous Enhancer Regions in Mouse and Human CCS

Distal EA1 binding sites include those that occurred at intragenic regions (36.5%) and intergenic regions (28.8%) (Figure 3A). These binding sites have an increased frequency of variant motifs relative to the canonical motif found most prevalently at promoter-associated binding sites (Figure 4B,E). Substantial overlap of binding sites and correlation of enrichments between EA1, RNAPOL2, and H3K27ac were retained when looking at distal regions only (Figure 6A,B). Enrichment of H3K27ac showed flanking bimodal peaks when centered at the EA1 binding sites, indicating EA1 occupancy of the intervening DNA and its facilitative role for these active regions (Figure 6B). This distribution of the H3K27ac signal differentiated putative enhancers from promoters, where H3K27ac enrichment was broader and directly overlapping (Figure 5B).

Intergenic super-enhancers (SEs) were identified by ROSE (Rank Ordering of Super Enhancers) using anti-H3K27ac ChIP-seq data in mCCS tumors (*n* = 4) [54,55]. The analysis revealed 584 SEs in mCCS (Figure 6C, top), 100% of which contained at least one EA1 binding site. EA1 ChIP-seq enrichment was strongest for binding sites that overlap with SEs (Figure 6D).

Anti-H3K27ac ChIP-seq in the SU-CCS-1 cell line, followed by the ROSE analysis, revealed 594 SEs in the human context of CCS (Figure 6C, bottom). In total, 161/594 (27%) contained at least one binding site for EA1 in the human intersectional EA1 ChIP-seq analysis. Again, a lower percentage in the human datasets was most likely a result of less efficient ChIP-seq methodology. Significant, upregulated genes were identified as adjacent to SEs in both species (Figure 6C, red), indicating the likely importance of the SE structure in regulating activated DEGs.

Mouse and human SEs were analyzed for synteny using the UCSC LiftOver tool. A total of 132 human SEs were syntenic to 126 mouse SEs and therefore were also associated with mouse EA1 ChIP-seq peaks (Figure 6E). These syntenic SEs may be facilitated by EA1 and represent the core regulatory chromatin circuitry of CCS. The nearest genes to these shared SEs were homologous genes in both species and were in the same orientation from the SE (Figure 6F). For example, *Nfil3* in mCCS had an SE 6.2 kb downstream, and *NFIL3* in SU-CCS-1 had an SE 5.9 kb downstream. These syntenic SEs, bound by EA1, were in the same orientation from homologous DEGs and may be important in CCS reprogramming, even if some were not identified in the intersectional method of EA1 ChIP-seq performed in human cell lines.

Anti-H3K27ac HiChIP chromatin conformation experiments that were reported for SU-CCS-1 were reanalyzed to identify looping between SEs and the promoters of genes being distally regulated [29]. A total of 594 human SEs looped to 3615 loci across the human genome, of which 1590 loci were at promoter regions of 746 unique genes. Interesting genes whose promoters are looped to SEs included *MITF*, *FUS*, *FOSL1*, *JUNB*, *TP53*, *SOX10*, and *TFAP2A*. In the SU-CCS-1 RNA-seq dataset, nineteen of these SE-looped genes had a log_2_FC value greater than 2 compared to three genes having a log_2_FC value less than −2 (Figure 6G). SE-looped genes were more likely to be transcriptionally active and represented how EA1 may be facilitating SEs to associate with and upregulate target genes through a chromatin conformational fashion.

## 4. Discussion

CCS was recapitulated in genetically engineered mice, which formed tumors in 1–2 months that modeled the human disease by morphology and transcriptome. These mice expressed V5-tagged *EA1*, allowing for robust ChIP-seq to probe where the fusion protein binds along the mouse genome. Expression of V5-tagged *EA1* to generate the tumor and utilization of an anti-V5 antibody rule out many antibody artifacts and tag-impingement problems that can occur with an in vivo analysis of transcription factors. These results corroborated findings from the human context, including a propensity to bind intragenic and intergenic regions. However, this analysis also identified novel binding motifs and many DEGs that were previously unreported for CCS.

We curated comprehensive lists of upregulated and downregulated target genes, whose promoters are bound directly by EA1 or are linked to distal EA1-bound enhancers. Differentially expressed genes being upregulated or downregulated by EA1 suggests secondary epigenetic mechanisms may determine transcriptional effects on target genes. For example, hypomethylation patterns correlate with activation of known target genes across the EWSR1::CREB family of tumors [46]. Methylation machinery at differentially expressed target genes may be dysregulated by EA1.

The analysis of recurrent motifs within EA1 binding sites revealed iterations of motifs that comprise TGA or the complementary TCA. We propose that EA1 has a novel propensity to bind TGA repeats at enhancers, similar to EWSR1::FLI1 binding of GGAA microsatellites in Ewing sarcoma [50]. Further research is required to understand the importance of these EA1-bound microsatellites in driving CCS. We also identified the AP1 motif as recurrent within EA1 binding sites, suggesting that AP1 factors may represent co-factors that associate with EA1 at these sites. AP1 factors, including FOS and JUN proteins, can form heterodimers with ATF factors and are known to facilitate enhancer regions by recruiting chromatin remodeling machinery [6,49,56].

The strongest promoter-associated EA1 binding site in mCCS occurred at the *Crem* promoter (Figure 5E). *CREM* may represent an important target gene of EA1 and possibly a co-factor with EA1. The occurrence of the *EWSR1::CREM* fusion in CCS, albeit rare, may indicate functionality of CREM in the presence of *EA1* expression [8]. Because *CREM* is a member of the ATF/CREB family, EA1 and CREM may dimerize with one another. Cofactor composition, whether involving AP1 or ATF/CREB factors, may dictate target gene reprogramming and require further investigation.

Lastly, SEs were independently identified in mouse and human CCS models and showed strikingly similar synteny when compared between these species. EA1 bound to 100% of SEs in mCCS. Significantly upregulated DEGs were identified nearby SEs in both species and showed 3D-looped association with SEs by HiChIP in human CCS. EA1 may be orchestrating the SE structure to regulate the epigenetic reprogramming of CCS.

## 5. Conclusions

Mouse modeling of CCS recapitulated morphology as well as transcriptional signatures including differentially expressed genes, fusion oncoprotein binding characteristics, and the super-enhancer structure. These experiments identified putative dependencies, such as important super-enhancers that associate with target genes, that can be tested in the mouse model moving forward. Epigenomic characterization of EA1 and transcriptomic analyses across mouse and human contexts provide the foundational framework for understanding the molecular underpinnings of CCS.

## Figures and Tables

**Figure 1 cancers-15-05750-f001:**
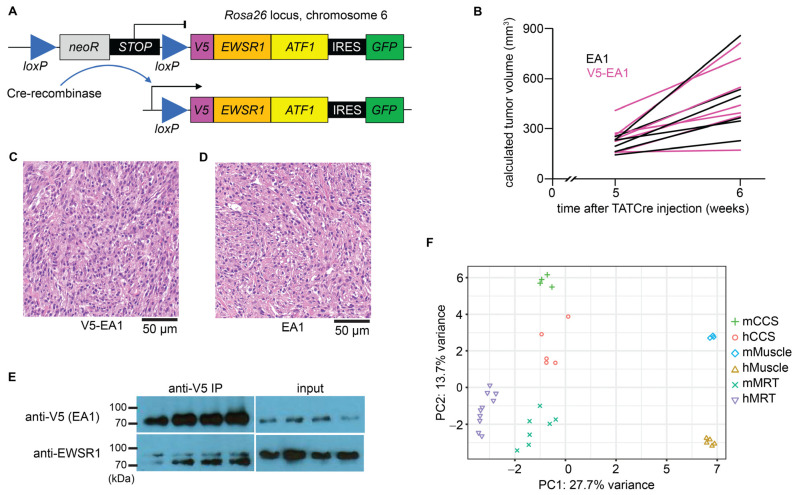
Conditional expression of V5-tagged EWSR1::ATF1 (EA1) drives tumorigenesis, which recapitulated the human CCS transcriptome. (**A**) Mouse cassette of inducible V5-tagged EWSR1::ATF1 cDNA engineered at the Rosa26 locus. (**B**) Growth rates of untagged EA1-expressing mouse tumors (black, *n* = 6) and V5-tagged EA1-expressing mouse tumors (pink, *n* = 8). (**C**) Histology of mouse model tumors expressing V5-tagged EA1. (**D**) Histology of mouse model tumors expressing untagged EA1. (**E**) Immunoblot of whole cell lysate derived from mCCS tumors (*n* = 4). Input lysate (**right**) was immunoprecipitated using an anti-V5 antibody (**left**). Immunoblot was performed using an anti-V5 antibody to identify V5-EA1 at approximately 80 kDa (**top**) and an anti-EWSR1 antibody to identify V5-EA1 and endogenous EWSR1 at approximately 95 kDa (**bottom**). (**F**) PCA plot depicting CCS mouse model tumors (mCCS), human CCS tumors (hCCS), mouse skeletal muscle (mMuscle), human skeletal muscle (hMuscle), MRT mouse model tumors (mMRT), and human MRT tumors (hMRT).

**Figure 2 cancers-15-05750-f002:**
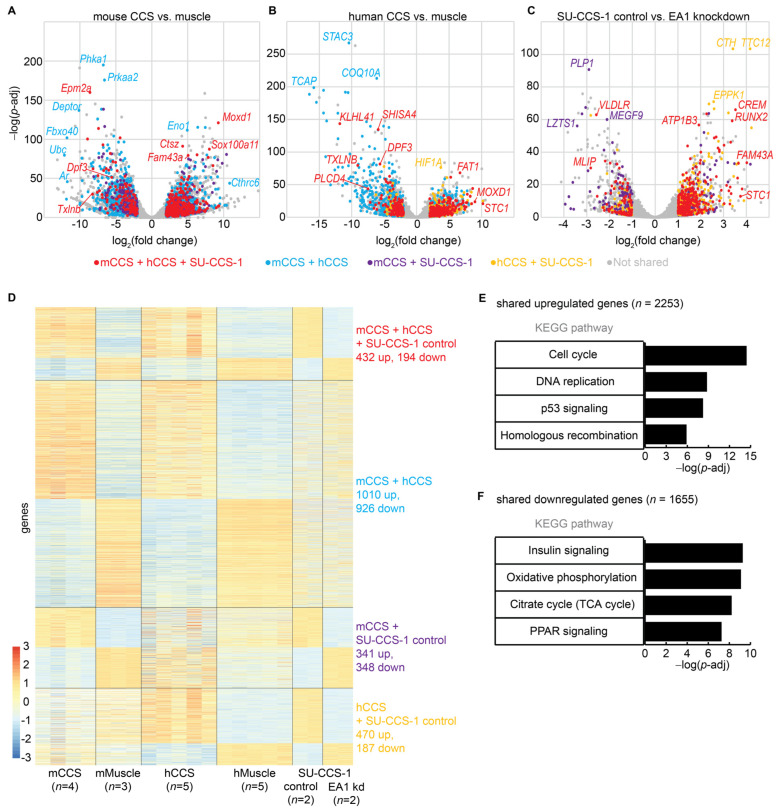
Transcriptomic analysis of mouse and human CCS identified shared differentially expressed genes involved in proliferation. (**A**) Volcano plot of gene expression data comparing TATCre-EA1 mouse model tumors vs. mouse skeletal muscle. (**B**) Volcano plot of gene expression data comparing human CCS tumors vs. human skeletal muscle. (**C**) Volcano plot of gene expression data comparing human SU-CCS-1 cell line, RNAi control vs. RNAi EA1 knockdown. (**D**) Heatmap of shared significant, differentially expressed genes, clustered by genes shared between datasets. (**E**) Gene set enrichment analysis of cellular pathways for genes that are upregulated in at least two RNA-seq datasets. (**F**) Gene set enrichment analysis of cellular pathways for genes that are downregulated in at least two RNA-seq datasets.

**Figure 3 cancers-15-05750-f003:**
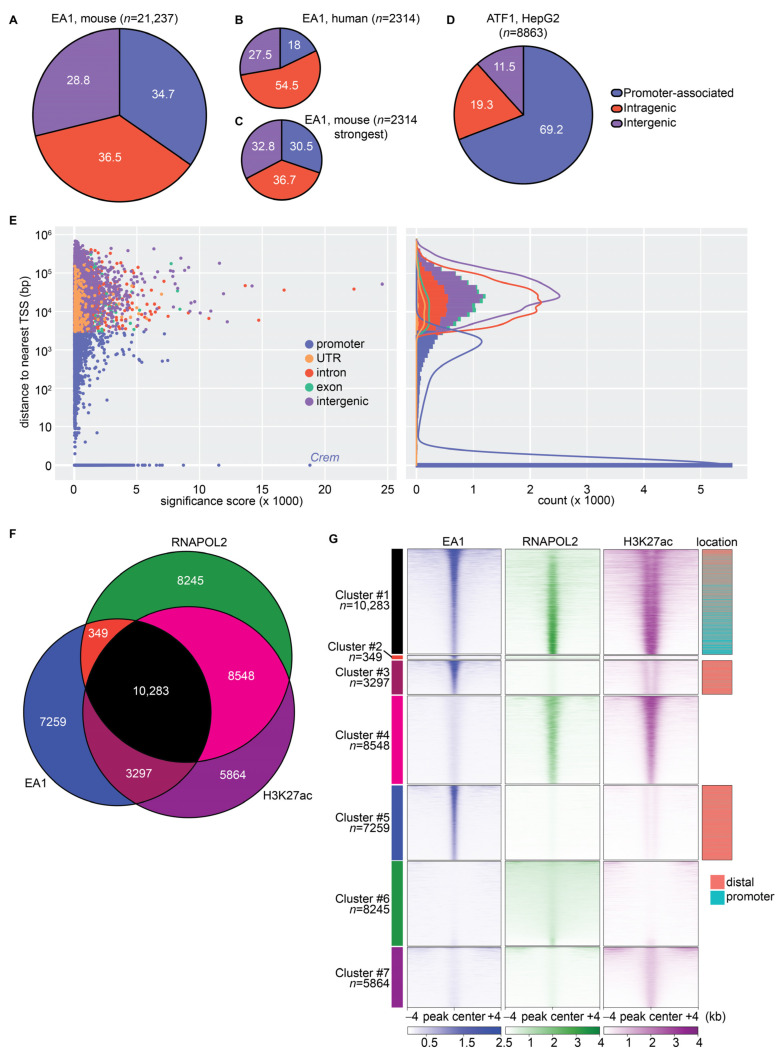
Genomic binding of EWSR1::ATF1 showed predilection of distal regions and correlation with RNAPOL2 and H3K27ac signatures. Genomic distributions of binding sites in ChIP-seq datasets: (**A**) total EA1 in mCCS, (**B**) total EA1 in human CCS cell lines, (**C**) high-stringency EA1 in mCCS, and (**D**) wtATF1 in HepG2 cell line. (**E**) (**Left**), scatter plot depicts all mCCS EA1 binding sites plotted by distance to nearest gene’s transcription start site (TSS) versus significance score. (**Right**), distribution of binding sites binned by distance to TSS versus total counts of binding sites within each bin. (**F**) Overlap of total binding sites between EA1, RNAPOL2, and H3K27ac ChIP-seq datasets. (**G**) Enrichment heatmap shows relative enrichments and shared clusters between EA1, RNAPOL2, and H3K27ac ChIP-seq datasets. Loci are centered on and ranked by EA1 enrichment in clusters #1, #2, #3, and #5. Loci are centered on and ranked by H3K27ac enrichment in clusters #4 and #7. Loci are centered on and ranked by RNAPOL2 enrichment in cluster #6. Called binding sites for EA1 are depicted as promoter-associated (teal) or distal (salmon) for clusters #1, #3, and #5.

**Figure 4 cancers-15-05750-f004:**
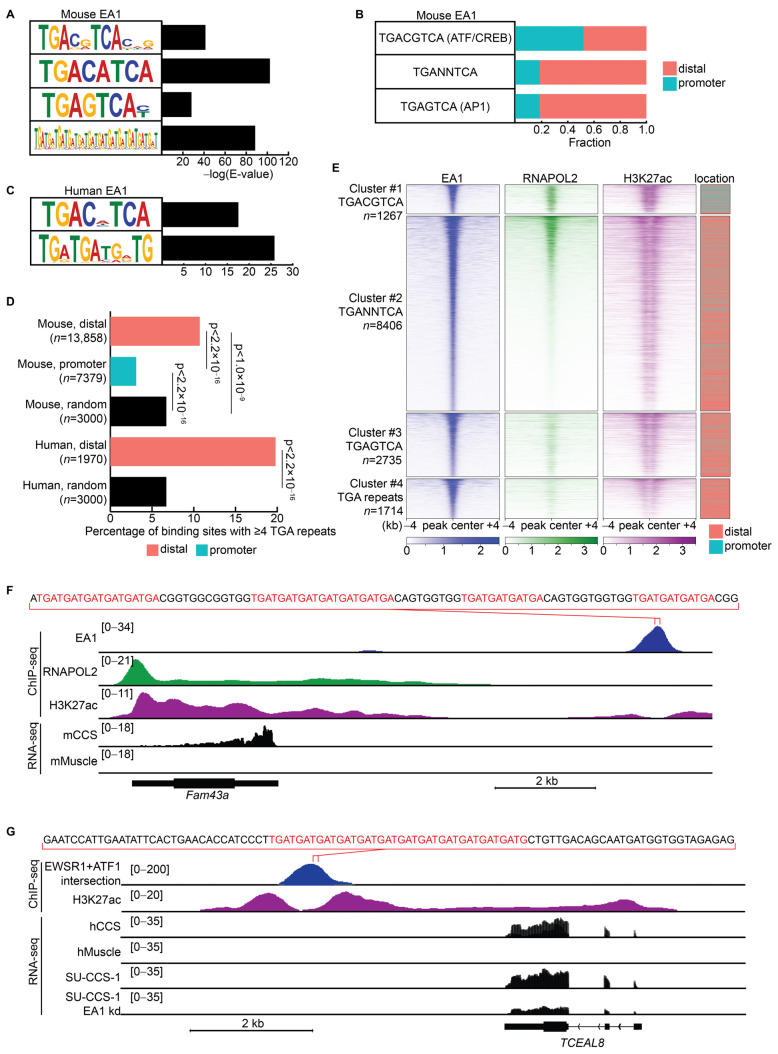
EWSR1::ATF1 bound the canonical ATF/CREB motif at promoters and novel variant motifs at distal regions. (**A**) Statistically significant, recurrent motifs identified in the center 100 base pairs of binding sites in the mCCS EA1 ChIP-seq dataset (*n* = 21,237). (**B**) Genomic distribution of binding sites that contain TGACGTCA (*n* = 1267), TGANNTCA (*n* = 7139, excluding TGACGTCA), or TGAGTCA (*n* = 2735), depicted as fractions of promoter-associated binding sites or distal binding sites. (**C**) Statistically significant, recurrent motifs identified in binding sites in the human CCS cell line EA1 ChIP-seq dataset (*n* = 2314). (**D**) Relative percentages of binding sites that contain TGA repeat motifs, as defined as 4 or more 3-base-pair motif strings with a tolerance gap of 2 base pairs. Statistically significant differences between datasets were calculated with unpaired *t*-test. (**E**) Enrichment heatmap shows relative enrichments between EA1, RNAPOL2, and H3K27ac ChIP-seq datasets clustered by the recurrent motif within the EA1 binding sites. Loci are centered on and ranked by EA1 enrichment. EA1 binding sites are depicted as promoter-associated (teal) or distal (salmon). (**F**) Example of a TGA-repeat containing mCCS EA1 binding site near *Fam43a*. Central 100 base pairs are displayed with TGA repeats highlighted in red. ChIP-seq reads per million (RPM) are depicted as colored tracks, and RNA-seq RPM are depicted in black. (**G**) Example of a TGA-repeat containing human CCS EA1 binding site near *TCEAL8*. Central 100 base pairs within the peak are displayed with TGA repeats highlighted in red. ChIP-seq RPM are depicted as colored tracks, and RNA-seq RPM are depicted as black tracks normalized to the same scale.

**Figure 5 cancers-15-05750-f005:**
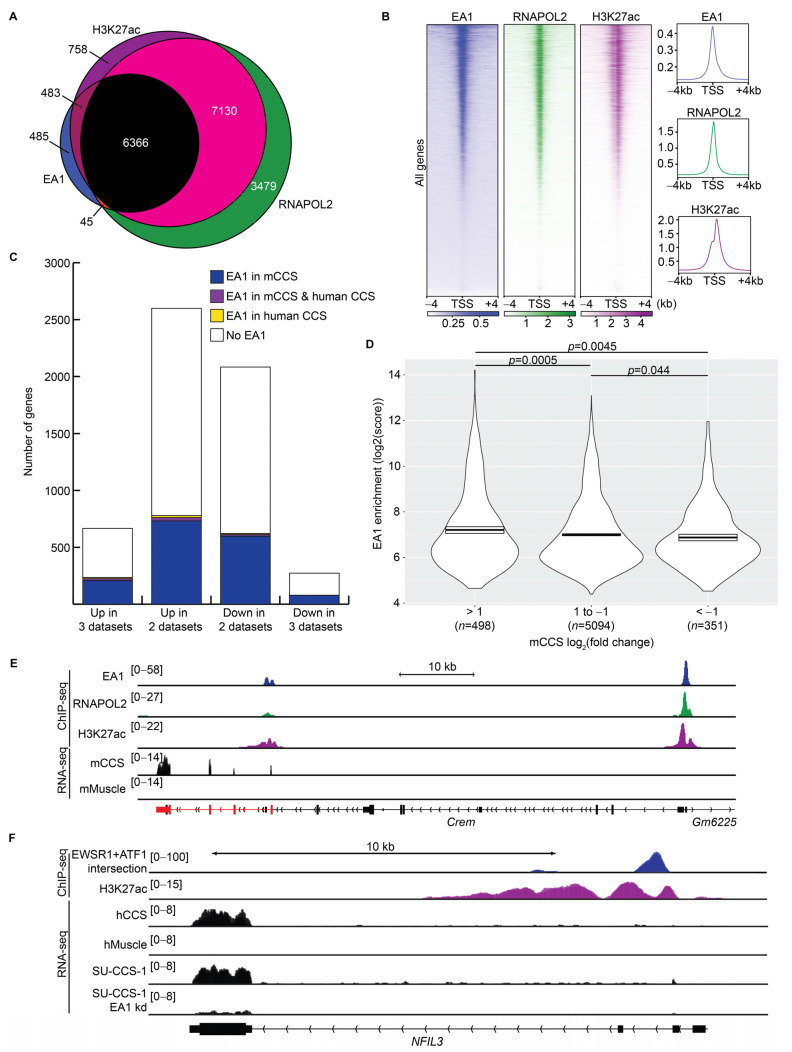
Promoter-associated EA1 binding sites promoted transcription of target genes. (**A**) Overlap of promoter-associated binding sites between EA1, RNAPOL2, and H3K27ac ChIP-seq datasets in mCCS. (**B**) ChIP-seq enrichment heatmaps (**left**) and averaged enrichments (**right**) of EA1, RNAPOL2, and H3K27ac at all mouse TSSs, centered on the TSSs and ranked by EA1 enrichment. (**C**) Overlayed bars represent the number of differentially expressed genes (upregulated in 3 datasets, upregulated in 2 datasets, downregulated in 2 datasets, or downregulated in 3 datasets) and are colored if the genes’ promoter regions (+/−3 kb) contain at least 1 EA1 binding site. (**D**) Enrichment of promoter-associated EA1 binding sites binned by bound genes’ log_2_FC values of >1, 1 to −1, or <−1. Statistically significant differences between datasets were calculated with unpaired *t*-test. (**E**) mCCS example of EA1 binding site at the *Crem* promoter. ChIP-seq reads per million (RPM) are depicted as colored tracks, and RNA-seq RPM are depicted in black. The highly-expressed isoform of *Crem* is depicted in red. (**F**) Human CCS example of EA1 binding site at the *NFIL3* promoter. ChIP-seq RPM are depicted as colored tracks, and RNA-seq RPM are depicted in black.

**Figure 6 cancers-15-05750-f006:**
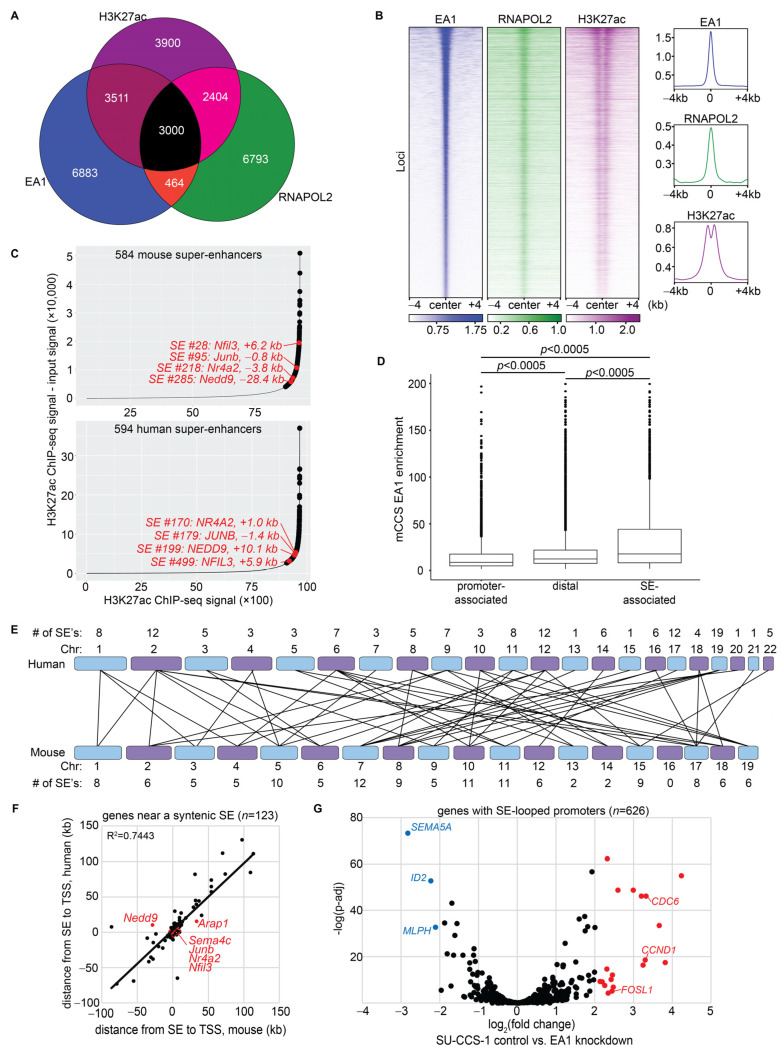
Distal EA1 binding sites demarcated homologous enhancer regions in mouse and human CCS. (**A**) Overlap of distal binding sites between EA1, H3K27ac, and RNAPOL2 ChIP-seq datasets in mCCS. (**B**) ChIP-seq enrichment heatmaps (**left**) and averaged enrichments (**right**) of EA1, RNAPOL2, and H3K27ac distal binding sites, centered on and ranked by EA1 enrichment. (**C**) ROSE plots identifying mouse super-enhancers (**top**) and human super-enhancers (**bottom**). Red labels show SEs in both species that associated with homologous genes upregulated in all three RNA-seq datasets. (**D**) Enrichment of mCCS EA1 between promoter-associated (*n* = 3257), distal (*n* = 7408), and SE-associated binding sites (*n* = 1920). Statistically significant differences between datasets were calculated by Tukey’s honestly significant difference test. (**E**) Synteny map depicting super-enhancers shared between SU-CCS-1 (**top**) and mCCS (**bottom**). (**F**) Genes plotted by the distance between their TSS and a nearby SE in mCCS (*x*-axis) and SU-CCS-1 (*y*-axis). Genes labeled in red are upregulated in all three RNA-seq datasets. (**G**) Volcano plot of gene expression data comparing human SU-CCS-1 cell line, RNAi control vs. RNAi EA1 knockdown, for genes whose promoters display HiChIP association with an SE. Genes labeled in blue have a log_2_FC value < −2, and genes labeled in red have a log_2_FC value > 2.

## Data Availability

All the raw and processed sequencing data were deposited on the Gene Expression Omnibus at GSE248753.

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
