# Peer review of "EWSR1::ATF1 Orchestrates the Clear Cell Sarcoma Transcriptome in Human Tumors and a Mouse Genetic Model"

_cancers, 2023, doi:10.3390/cancers15245750_

Round 1

Reviewer 1 Report

Comments and Suggestions for Authors

In this manuscript, the authors provide an interesting study on the EWSR1::ATF1 fusion, thus potentially benefiting future clear cell sarcoma (CCS) studies and patients. Yet, some additional elaboration and data would be really appreciated. See more details below.

It is suggested to provide some detailed information in the Materials and Methods section, such as the mouse strain used and its resources, the total number of mice used, the number of mouse model tumors used in the immunochemistry analysis and ChIP-sequencing, the method of calculating the tumor volumes, the mouse sacrifice timing, and statistical analysis methods and software, etc. Concerning the algorithm developed by the authors, will it be made available to the public, such as on GitHub? Figures 1C and 1D provide HE staining results, but I am not sure it is covered in the Materials and Methods section.

Concerning Figure 1E, is there any special reason why the input was not adjusted to the same level? What was the antibody used in the input blot? Concerning Figure 1F and its relevant descriptions, it would be interesting to compare mouse model tumors from the tagged V5-EA1 group vs. the untagged EA1 group, if possible. I believe this will better support the conclusion drawn there.

Concerning Figure 2D and its relevant descriptions, I assume the SU-CCS-1 mark on the right-hand side refers to SU-CCS-1 (control). Is it possible to make it clear in the figure? Also, the genes shared among mMuscle, hMuscle and SU-CCS-1 (EA1 kd) are not included in the counting according to the figure. Along this line, is it accurate to say “2,253 upregulated genes and 1,655 downregulated genes were identified across two or more datasets” (emphasis added)? Further, similar to the comment above, is it possible to compare the genes having an altered expression between the tagged V5-EA1 group and the untagged EA1 group?

Concerning Figure 3 and its relevant descriptions, it seems the sentence starting in line 324 does not belong to the figure legend. Also, the sentence in line 352 seems incomplete.

Concerning Figure 4 and its relevant descriptions, it seems the break between lines 408 and 409 is not necessary.

Concerning Figure 5 and its relevant descriptions, is the effect observable at the protein/translational level?

Also, it is suggested to provide the full names of acronyms upon their first appearance and use the acronyms from there, for example, MRT, mCCS, and DEGs.

Author Response

Reviewer #1. In this manuscript, the authors provide an interesting study on the EWSR1::ATF1 fusion, thus potentially benefiting future clear cell sarcoma (CCS) studies and patients. Yet, some additional elaboration and data would be really appreciated. See more details below.

It is suggested to provide some detailed information in the Materials and Methods section, such as the mouse strain used and its resources, the total number of mice used, the number of mouse model tumors used in the immunochemistry analysis and ChIP-sequencing, the method of calculating the tumor volumes, the mouse sacrifice timing, and statistical analysis methods and software, etc. Concerning the algorithm developed by the authors, will it be made available to the public, such as on GitHub?  

We added: "The mouse strain background was 129/SvJ and C57Bl/6 as previously described [5]."

We added "Four..." to the beginning of the line describing how many tumors were processed, initially.

We added the words "5 samples with", "5 samples with", and "4 samples with" inside the parentheses on lines 157-158

We added: "...was measured using calipers and calculated using an ellipsoidal formula: volume = (length * width * width) * 0.5."

We edited the sentence: "The protocol under which this mouse strain was generated, these mouse tumors were grown, and euthanasia criteria and timing were described was protocol #16-10012."

For the standard analysis of RNASeq and ChIPSeq data, including differential expression analysis and peak calling, we employed software that is comprehensively described in the Materials and Methods section. Genomic visualizations and ChiPSeq heatmaps were primarily created using the Integrative Genomics Viewer (IGV) and Deeptools respectively, details of which are also provided in the Materials and Methods. For additional graphical representations, we utilized R programming throughout this study. The bespoke algorithm developed for motif scanning has been made publicly accessible via our GitHub repository: GitHub Link. Given that the method is a specialized solution rather than a full-fledged package, it has been shared within an existing repository rather than establishing a separate one.

Figures 1C and 1D provide HE staining results, but I am not sure it is covered in the Materials and Methods section.

We added a description of the H&E sample processing as Materials and Methods section 2.9.

Concerning Figure 1E, is there any special reason why the input was not adjusted to the same level? 

This is certainly not the most beautiful blot, but it sufficed to prove the point that the protein was expressed, so we did not re-balance well loading for a better image.

What was the antibody used in the input blot?

We added a phrase that indicates that the same antibody was used on the immunoprecipitated and input samples.

Concerning Figure 1F and its relevant descriptions, it would be interesting to compare mouse model tumors from the tagged V5-EA1 group vs. the untagged EA1 group, if possible. I believe this will better support the conclusion drawn there.

These were from tagged V5-EA1.

Concerning Figure 2D and its relevant descriptions, I assume the SU-CCS-1 mark on the right-hand side refers to SU-CCS-1 (control). Is it possible to make it clear in the figure? 

We added these “control” labels to the figure panel.

Also, the genes shared among mMuscle, hMuscle and SU-CCS-1 (EA1 kd) are not included in the counting according to the figure. 

Correct, the shared genes mMuscle, hMuscle, and SU-CCS-1 EA1 kd were not identified themselves. However, our differential expression analysis likely identified such genes as indicated visually in the figure.

Along this line, is it accurate to say “2,253 upregulated genes and 1,655 downregulated genes were identified across two or more datasets” (emphasis added)? 

The word "CCS" has been added to these sentences for clarity.

Further, similar to the comment above, is it possible to compare the genes having an altered expression between the tagged V5-EA1 group and the untagged EA1 group?

We have separately performed RNA-seq for each and found the only differences between the two to relate to batch effects.

Concerning Figure 3 and its relevant descriptions, it seems the sentence starting in line 324 does not belong to the figure legend. 

Also, the sentence in line 352 seems incomplete.

This formatting mistake has been corrected.

Concerning Figure 4 and its relevant descriptions, it seems the break between lines 408 and 409 is not necessary.

This has been corrected

Concerning Figure 5 and its relevant descriptions, is the effect observable at the protein/translational level?

The levels of CREM & NFIL3 proteins were not tested. CREM protein, in particular, is hypothesized to be important in the transcriptional programming of CCS and is a top candidate for future studies.

Also, it is suggested to provide the full names of acronyms upon their first appearance and use the acronyms from there, for example, MRT, mCCS, and DEGs.

Each acronym is now named fully prior to the first parenthetical mention.

Reviewer 2 Report

Comments and Suggestions for Authors

Thank you for giving me the chance to review this interesting paper.

The study presents comprehensive research on how the V5-tagged EWSR1::ATF1 fusion oncoprotein is distributed in clear cell sarcoma at a genomic level. The introduction successfully establishes the significance of comprehending the distribution pattern of this fusion oncoprotein, given its rarity and aggressive nature in CCS cases. The methodology, involving the expression of V5-tagged EWSR1::ATF1 in mice to induce tumors that recapitulate the human CCS transcriptome, seems appropriate for addressing the research question.

The use of ChIP-sequencing for both V5-EWSR1::ATF1 and H3K27ac provides a comprehensive analysis of the genomic landscape.
The discovery of previously unidentified patterns, such as the AP1 motif and the motif composed of TGA repeats, brings a fresh perspective to the research.

Moreover, identifying shared super-enhancer structures linked to active genes in both mouse models and human CCS contexts is a noteworthy breakthrough.
I believe that this translational research has the potential to offer important insights into the clinical implications and treatment possibilities based on the molecular characteristics that have been identified.

I suggest accepting it for publication in present form.

Author Response

Thank you for the kind review.

Reviewer 3 Report

Comments and Suggestions for Authors

The authors presented an accurate analysis of V5-tagged EWSR1:ATF1 fusion protein genomic occupancy in mouse CCS models and in human CCS cell lines. Data of bulk RNA-Seq in CCS samples and mouse tissues are also reported and integrated.

The overall study is well-designed and the results are exhaustively described. I have only a few minor comments that the author should address to improve their manuscript:

- Please indicate the GEO identifiers of each analyzed dataset

- Integration with independent CCS data will be valuable. For example, any noncoding genetic variants associated with CCS are mapped within the identified E41 binding sites?

Author Response

    - Please indicate the GEO identifiers of each analyzed dataset

All of the data have been deposited on GEO, in super series GSE248753. The identifier is now listed in the data availability statement.

    - Integration with independent CCS data will be valuable. For example, any noncoding genetic variants associated with CCS are mapped within the identified E41 binding sites?

We look forward to doing this work in the near future.

Round 2

Reviewer 1 Report

Comments and Suggestions for Authors

Thanks for the authors’ amendments and elaboration. I have no further comments.